# Sex-specific differences in physiological parameters related to SARS-CoV-2 infections among a national cohort (COVI-GAPP study)

Kirsten Grossmann[1,2☯], Martin Risch[2,3,4☯], Andjela Markovic[5,6,7☯], Stefanie Aeschbacher[8], Ornella C. Weideli[2,9], Laura Velez[2], Marc Kovac[4], Fiona Pereira[10], Nadia Wohlwend[4], Corina Risch[4], Dorothea Hillmann[4], Thomas Lung[4], Harald Renz[11], Raphael Twerenbold[8,12], Martina Rothenbühler[5], Daniel Leibovitz[5], Vladimir Kovacevic[5], Paul Klaver[13], Timo B. Brakenhoff[13], Billy Franks[13], Marianna Mitratza[14,15], George S. Downward[14,15], Ariel Dowling[16], Santiago Montes[17], Duco Veen[18,19], Diederick E. Grobbee[14,15‡], Maureen Cronin[5‡], David Conen[20‡], Brianna M. Goodale[5,13‡], Lorenz Risch[1,2,4,21‡*], on behalf of the COVID-19 remote early detection (COVID-RED) consortium[¶]

1 Private University in the Principality of Liechtenstein (UFL), Triesen, Principality of Liechtenstein, 2 Dr Risch Medical Laboratory, Vaduz, Liechtenstein, 3 Central Laboratory, Kantonsspital Graubünden, Chur, Switzerland, 4 Dr Risch Medical Laboratory, Buchs, Switzerland, 5 Ava AG, Zürich, Switzerland, 6 Department of Psychology, University of Fribourg, Fribourg, Switzerland, 7 Department of Pulmonology, University Hospital Zurich, Zurich, Switzerland, 8 Cardiovascular Research Institute Basel (CRIB), University Hospital Basel, University of Basel, Basel, Switzerland, 9 Soneva Fushi, Boduthakurufaanu Magu, Male, Maldives, 10 Department of Metabolism, Digestive Diseases and Reproduction, Imperial College London, South Kensington Campus, London, United Kingdom, 11 Institute of Laboratory Medicine and Pathobiochemistry, Molecular Diagnostics, Philipps University Marburg, Marburg, Germany, 12 Department of Cardiology and University Center of Cardiovascular Science, University Heart and Vascular Center Hamburg, Hamburg, Germany, 13 Julius Clinical, Zeist, The Netherlands, 14 UMC Utrecht, Utrecht, The Netherlands, 15 Julius Global Health, the Julius Center for Health Sciences and Primary Care, University Medical Center, Utrecht, The Netherlands, 16 Takeda Pharmaceuticals, Digital Clinical Devices, Cambridge, Massachusetts, United States of America, 17 Roche Diagnostics Nederland B.V., Almere, The Netherlands, 18 Department of Methodology and Statistics, Utrecht University, Utrecht, The Netherlands, 19 Optentia Research Programme, North-West University, Potchefstroom, South Africa, 20 Population Health Research Institute, McMaster University, Hamilton, Canada, 21 Center of Laboratory Medicine, University Institute of Clinical Chemistry, University of Bern, Inselspital, Bern, Switzerland

☯ These authors contributed equally to this work.
‡ DEG, MC, DC, BMG and LR also contributed equally to this work.
¶ Full list of group members of the COVID-RED consortium is provided in the acknowledgments.
* lorenz.risch@ufli.li

**Data Availability Statement:** An anonymized version of the datasets generated in the COVID-RED trial as well as supporting documentation are

## Abstract

Considering sex as a biological variable in modern digital health solutions, we investigated sex-specific differences in the trajectory of four physiological parameters across a COVID-19 infection. A wearable medical device measured breathing rate, heart rate, heart rate variability, and wrist skin temperature in 1163 participants (mean age = 44.1 years, standard deviation [SD] = 5.6; 667 [57%] females). Participants reported daily symptoms and confounders in a complementary app. A machine learning algorithm retrospectively ingested daily biophysical parameters to detect COVID-19 infections. COVID-19 serology samples were collected from all participants at baseline and follow-up. We analysed potential sex-specific differences in physiology and antibody titres using multilevel modelling and t-tests. Over 1.5 million hours of physiological data were recorded. During the symptomatic period

publicly available through DataverseNL at the following link (doi:10.34894/FW9P07). Further data that underlie the results reported in this paper were collected from study participants from the Principality of Liechtenstein, a very small country, where the risk of subject identification is increased due to the size of the population (less than 40'000 inhabitants). To respect data protection and to prevent the identification of participants, data access is restricted to researchers meeting the criteria for access to confidential data. Data are available from (contact: lorenz.risch@ufl.li, martin.risch@ksgr.ch, and david.conen@phri.ca). Further, the data underlying the results presented in the study are available from (Private University of the Principality of Liechtenstein, Institutional Review Board, 9495 Triesen; irb@ufl.li).

**Funding:** This work has received support from the Princely House of the Principality of Liechtenstein, the government of the Principality of Liechtenstein, the Hanela Foundation in Switzerland, and the Innovative Medicines Initiative (IMI) 2 Joint Undertaking under grant agreement No 101005177. This Joint Undertaking receives support from the European Union's Horizon 2020 research and innovation programme and EFPIA. The funders had no role in study design, data collection and analysis, decision to publish, or preparation of the manuscript.

**Competing interests:** The authors have read the journal's policy and have the following competing interests: Lorenz Risch, and Martin Risch are key shareholders of the Dr Risch Medical Laboratory. David Conen has received consulting fees from Roche Diagnostics, outside of the current work. Andjela Markovic, Vladimir Kovacevic, Martina Rothenbühler, Brianna Goodale and Maureen Cronin are past employees of Ava AG. Brianna Goodale and Timo Brakenhoff are current employees of Julius Clinical BV. Billy Franks is a former employee of Julius Clinical BV and now an employee of Haleon. Paul Klaver and Duco Veen are former employees of Julius Clinical BV. Marianna Mitratza is a current employee of P95 CVBA. There are no patents, products in development or marketed products associated with this research to declare. These competing interests do not alter our adherence to PLOS ONE policies on sharing data and materials.

of infection, men demonstrated larger increases in skin temperature, breathing rate, and heart rate as well as larger decreases in heart rate variability than women. The COVID-19 infection detection algorithm performed similarly well for men and women. Our study belongs to the first research to provide evidence for differential physiological responses to COVID-19 between females and males, highlighting the potential of wearable technology to inform future precision medicine approaches.

## Introduction

On March 11, 2020, the WHO declared the fast-spreading coronavirus disease (COVID-19) a global pandemic [1]. This novel viral disease was first detected in Wuhan, China, in December 2019 and is caused by severe acute respiratory syndrome coronavirus (SARS-CoV-2) [2]. Increasing knowledge about risk factors and symptoms, as well as the implementation of mass reverse transcription polymerase chain reaction (RT-PCR), serological tests, vaccines, and social restrictions have helped control its spread [3,4]. However, asymptomatic virus transmissions and emerging virus mutations pose ongoing challenges in dealing with the pandemic. Today, more than two years after the first case was detected, many countries worldwide continue to experience waves of rising infections, with numerous unknowns remaining in our understanding of SARS-CoV-2. In particular, consistent data about the role of sex in relation to COVID-19 are lacking [5,6]. Significant changes in physiological parameters such as breathing rate, heart rate, heart rate variability, and wrist skin temperature during a COVID-19 infection [7] raise the question about sex-specific differences within the trajectory of these parameters. A better understanding of sex-specific trajectories in physiological responses to the infection may support early detection and treatment of COVID-19.

A meta-analysis found that men with COVID-19 were globally almost three times more likely than women to be admitted to an intensive treatment unit [8]. Furthermore, the disease's mortality rates were higher in men [9], potentially due to sex-specific differences in angiotensin-converting enzyme 2 (ACE2) expression [10,11]. On the other hand, women were found to more frequently experience persistent symptoms such as dyspnoea and fatigue several months after the acute phase of the illness [12]. The infection rates were similar between the sexes [8], although this observation may differ between countries [13]. Moreover, initial analyses of eumenorrheic women's susceptibility to SARS-CoV-2 among a real-world sample are in line with previously shown immune function fluctuations across the menstrual cycle [14] and suggest increased susceptibility during the luteal phase [15]. Research on sex-specific differences in immune responses that underlie COVID-19 disease outcomes showed higher plasma levels of innate immune cytokines such as IL-8 and IL-18 along with more robust induction of non-classical monocytes in male patients, whereas female patients showed higher T cell activation during SARS-CoV-2 infection [16]. Also, higher levels of innate immune cytokines were associated with worse disease progression in female patients [16].

Previous studies have shown that direct-to-consumer and easy-to-use products with wide market availability, such as Fitbit [17], smartwatches [18], the Ava bracelet [7,19], and other wearable devices [20] could be used for surveillance of changes in physiological parameters to give the user an early warning before COVID-19 symptom occurrence [21] or during asymptomatic infection [22]. The COVI-GAPP study investigated the applicability of the Ava bracelet for pre-symptomatic detection of COVID-19 [23]. Developed as a fertility tracker, the bracelet measures physiological parameters, including wrist skin temperature, breathing rate,

heart rate, heart rate variability, and skin perfusion [24]. The previously published interim analysis of the COVI-GAPP dataset demonstrated significant changes in skin temperature, breathing rate, heart rate, and heart rate variability during a COVID-19 infection [7]. These parameters were used to develop a machine learning (ML) algorithm for the detection of pre-symptomatic SARS-CoV-2 infection, which successfully detected 68% of COVID-19 cases up to two days before symptom onset. The algorithm is currently being tested and validated in a larger population with real-time access to the algorithm's predictions [19].

The current work analyzed the same physiological parameters collected in the COVI-GAPP study to quantify sex-specific differences before, during, and after a COVID-19 infection. We examined differences in trajectories of physiological parameters over five defined phases (baseline, incubation, pre-symptomatic, symptomatic, and recovery) between female and male participants. Furthermore, we evaluated the performance of our ML algorithm for female and male participants separately with the goal of assessing and correcting a potential sex bias in its functionality. Finally, we examined sex differences in antibody levels following COVID-19 to gain additional insights into sex-specific immune responses.

## Materials and methods

The current study was based on the COVI-GAPP research initiative and included continuous monitoring of biophysical signals by means of a wearable device, the Ava bracelet, coupled with periodic blood tests to assess SARS-CoV-2 antibody titres. Additionally, a ML algorithm was developed based on the COVI-GAPP data to aid in the early detection of COVID-19. This section provides an overview of the methodology employed to address the study's three primary objectives: 1) investigation of sex differences in COVID-19-related physiological parameters; 2) evaluation of the sex-specific performance of a ML algorithm for early COVID-19 detection; and 3) analysis of sex differences in antibody titres following COVID-19.

### Study design and participants

Since 2010, the observational population-based Genetic and Phenotypic Determinants of Blood Pressure and Other Cardiovascular Risk Factors (GAPP) study aims to better understand the development of cardiovascular risk factors in the general population of healthy adults aged 25 to 41 years [25]. From 2170 GAPP participants, 1163 individuals were enrolled in the COVI-GAPP study with inclusion and exclusion criteria published previously [23]. Data were collected from April 14, 2020, until January 31, 2022. The local ethics committee (KEK, Zürich, Switzerland) approved the study protocol, and written informed consent was obtained from each participant prior to enrolment (BASEC 2020–00786).

### Data collection

**1. Ava bracelet.**   Physiological parameters of interest for this analysis were breathing rate, heart rate, heart rate variability, and wrist skin temperature. They were measured every 10 seconds by a wrist-worn bracelet while the user slept. If a minimum of 4 hours of relatively uninterrupted sleep is achieved, proprietary manufacturer algorithms are employed for pre-processing to eliminate artifacts, identify sleep stages, and provide the nightly physiological parameters. To mitigate potential fluctuations during transitions between wakefulness and sleep, the initial 90 and final 30 minutes of data from each night were excluded. Additionally, each physiological parameter underwent locally estimated scatterplot smoothing (LOESS) before analysis to reduce artificial fluctuations due to measurement errors, aligning with previously established best practices [26]. Further details on the applied data cleaning practices described by the manufacturer can be found in previous publications [7,27].

The CE-certified and FDA-cleared Ava Fertility Tracker (version 2.0; Ava AG, Switzerland) was originally built to detect ovulating women's fertile days in real time with 90% accuracy [27–29]. The bracelet's three sensors can track biophysical changes regardless of the wearer's sex [7]. In the current study, they were used for detecting infection-based deviations from baseline parameters in both men and women (regardless of their menstruating status). In order to meet the European Union's General Data Protection Regulation (GDPR) requirements on participant data, log-in procedure and data handling were performed with an anonymized email account. Participants synchronized their bracelets each morning upon waking to a complementary smartphone app. Participants were reviewed by a weekly compliance report showing synchronization rates. The study team contacted individuals to follow-up with log-in issues or operational challenges, therefore ensuring quality control.

In addition to automatically collected physiological data, participants also provided information in the complementary app about their daily alcohol, medication, and drug intake (for more information see Risch et al. [7]), as these substances can alter central nervous system functioning [30]. Furthermore, the app collected information about comorbidities that could potentially influence the physiological signals. Finally, the app provided a customized user functionality where participants reported COVID-19 symptoms in a daily diary. Participants were also able to see and monitor changes in their physiological parameters in the app.

**2. SARS-CoV-2 antibody testing.** SARS-CoV-2 antibody tests were performed by the medical laboratory Dr Risch Ostschweiz AG (Buchs SG, Switzerland) with an orthogonal test algorithm employing electrochemiluminescence (ECLIA) assays testing for pan-immunoglobulins directed against the N antigen (sensitivity of 96%, specificity of 99.9% for recognition of past SARS-CoV2 infection) and the receptor binding domain (RBD) of the SARS-CoV-2 spike protein (sensitivity of 97.6%, specificity of 99.8% for recognition of past SARS-CoV2 infection), as described by Schaffner et al. [31] and Weber et al. [32]. The enacted procedure ensures testing for actual SARS-CoV-2 infection independent of vaccine status. Baseline data were collected starting in April 2020 onwards (run 1; R1). Three follow-up blood samples (run 2, R2; run 3, R3; and run 4, R4) were collected within the scope of the study (Fig 1). The cut-off levels used for positive and negative values were $\geq 1.0$ and $\leq 0.1$, respectively. Values between 0.2–0.9 were considered as gray zone. Seroconversion was assumed if the first blood sample was negative for SARS-CoV-2 antibodies but a subsequent sample was positive. Follow-up calls with participants who tested positive were performed to discuss their symptoms and duration.

**3. Questionnaires.** When visiting the study centre for SARS-CoV-2 antibody tests, participants were asked to answer a questionnaire about their personal information (age, sex), smoking status (current, past, never), as well as symptoms and hospitalizations during COVID-19 infection. These visits occurred at approximately 6-month intervals across the duration of data collection. Body mass index (BMI) based on height and weight was calculated with data from the GAPP database.

## Statistical analysis

Our primary objective was to examine sex differences in the trajectory of daily levels of the four physiological parameters across a SARS-CoV-2 infection (i.e., breathing rate in breaths per minute, skin temperature in degree Celsius, heart rate in beats per minute, heart rate variability). Heart rate variability was quantified as the ratio of low-frequency (0.04–0.15 Hz) to high-frequency (0.15–0.4 Hz) oscillations, as previously described [7]. Secondarily, we evaluated a machine learning algorithm designed for early detection of COVID-19 separately in male and female participants to examine potential sex biases in algorithm performance.

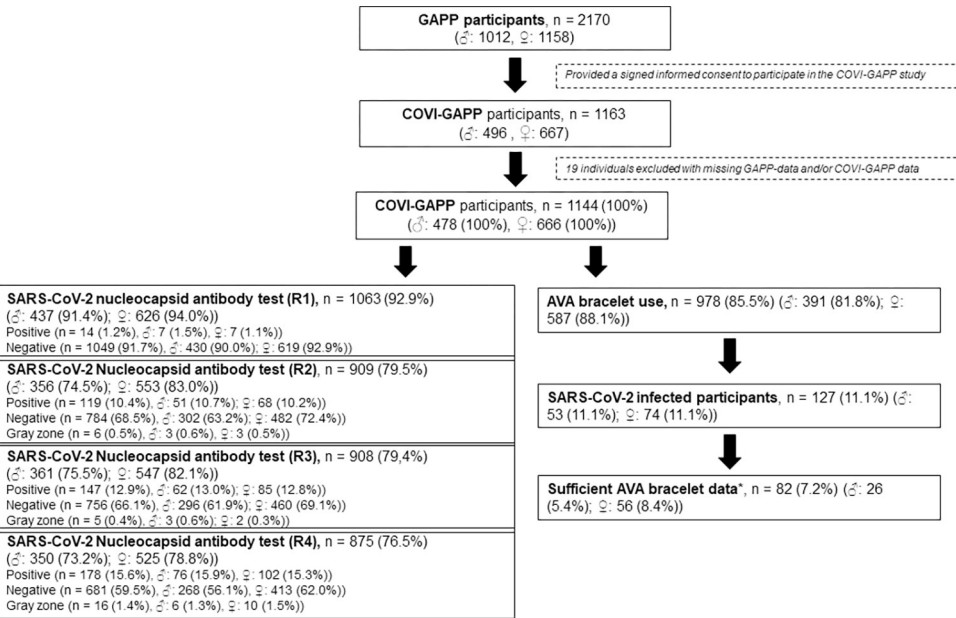

**Fig 1. Study flow chart of the 1,163 participants that are enrolled in the COVI-GAPP study.** The cut-off levels used for positive and negative values were ≥ 1.0 and ≤ 0.1, respectively. Values between 0.2–0.9 were considered as gray zone * Successful bracelet synchronization on more than 50% of days around symptom onset.

Finally, we assessed sex-specific differences in antibody titres after SARS-CoV-2 infections. We processed and analysed all data using R (version 4.1.1) [33] and Python (version 3.6) [34].

**1. Sex-specific differences in COVID-19 related physiological parameters.** To examine the association between sex and physiological parameters during baseline, incubation, pre-symptomatic, symptomatic, and recovery phases of a COVID-19 infection, we applied multi-level linear mixed models with random intercepts and slopes including residual maximum likelihood estimation (REML) and Satterthwaite degrees of freedom. A multiplicative interaction term tested the association between sex and the infection phase. All signals measured more than 10 days before symptom onset via phone call confirmation with a study team member were categorized as occurring during the baseline period. The incubation period was defined as the time interval from 10 days up to 3 days before symptom onset. The pre-symptomatic period was defined as the two days before symptom onset, while the symptomatic period lasted from the day of symptom onset until the day symptoms ended. All signals measured after symptom end were categorized as occurring during the recovery period. We dummy-coded four variables to indicate the period within which the signal occurred, with the baseline serving as the reference period. Each of the four multilevel models was compared to the corresponding null model (i.e., an intercept-only model) by means of an ANOVA.

**2. Sex-specific differences in algorithm's performance.** The retrospective ML algorithm, developed as described in previous papers [7,19] aimed to detect a COVID-19 infection prior to symptom onset. The algorithm was designed to ingest trends in physiological signals across sets of days to detect deviations in these signals and predict a potential infection. The model was trained to predict infection two days and one day prior to symptom onset, as well as on the day of symptom onset. Here, we assessed the algorithm's performance metrics separately for males and females to identify any potential sex bias in the model. Performance metrics were calculated per day in participants who tested positive, where days from -40 to -2 relative to the onset of the first symptoms were considered negative and days from -2 to day 0 as

positive. In other words, positive predictions of the algorithm prior to 2 days before symptom onset were interpreted as false positives. The set of metrics selected for the evaluation of the algorithm included precision (the number of true positives divided by the sum of true positives and false positives), recall (the number of true positives divided by the sum of true positives and false negatives), and F-score (the harmonic mean of precision and recall).

**3. Sex-specific differences in antibody titres of SARS-CoV-2 Nucleocapsid after COVID-19 infection.** To gain a deeper understanding of sex-specific differences in the immune system's reaction to the virus, antibody trajectories were monitored during the study period. Antibody titres reflected the concentration of antibodies in the blood that are specific to the SARS-CoV-2 virus. To enable a reliable comparison of antibody titres after a COVID-19 infection, antibody titres (values > 1.0) against the SARS-CoV-2 Nucleocapsid were compared between the sexes. Blood was collected four times over the course of the study with varying sample sizes (Fig 1). Normally distributed variables were compared using unpaired t-tests, and non-normally distributed variables were compared using Mann-Whitney U tests.

## Results

### Participants

A total of 1163 participants (mean age = 44.1 years, standard deviation [SD] = 5.6; 667 [57%] females) were enrolled in the study. During the study period, 127 participants (10.9%; [9.3,12.8]) contracted COVID-19. Eighty-two participants (mean age = 42.6 ± 5.3 years; 56 [68%] females) testing positive for SARS-CoV-2 had worn and synchronized their bracelet successfully on more than 50% of days around symptom onset (i.e., at least 20 days before and 20 days after symptom onset), thereby ensuring sufficient quality of data to be included in analyses. The number of days with successfully synchronized bracelet data did not differ ($p$ = 0.967) between females (range 67 to 511 days; mean = 239.6 ± 71.8 days) and males (range 45 to 508 days; mean = 238.8 ± 86.4 days). With regards to the reported symptom duration, values for four participants (2 females) were missing and imputed based on the median across the sample.

Blood samples and questionnaire data were available from 1,144 participants. The mean age and BMI of these participants were 45 (± 5.5) and 24.7 (± 3.9), respectively. At baseline, male participants had significantly higher BMIs (26.17 ± 3.41) than female participants (23.70 ± 3.96; t(1079) = 10.71, $p$<0.001). They also reported significantly higher rates of hypertension (7.74%) than female participants (3.15%; $X^2$(1) = 11.23, $p$<0.001). Analyses did not reveal any significant sex-based differences in smoking status, age, or hospitalization rate (Table 1).

### Sex-specific differences in COVID-19 related physiological parameters

We show the trajectory of each of the four analysed physiological parameters during a SARS-CoV-2 infection separated by sex (Fig 2). The multilevel models revealed significant differences between male and female participants in all parameters during the symptomatic period (Table 2). We observed a larger increase in skin temperature, breathing rate, and heart rate, as well as a larger decrease in heart rate variability in males compared to females during this period. Moreover, male participants' breathing rate and heart rate remained at significantly higher levels during the recovery period as compared to their female peers (Table 2). Each of the four models provided a significantly better fit to the data than the corresponding null model (p<0.0001).

As a sensitivity analysis, we also tested potentially confounding variables as single terms in additional models to determine whether changes in physiological parameters occurred due to

**Table 1. Sex differences in baseline characteristics.**

| Variables | Total<br>n = 1,144 | Male<br>n = 478 | Female<br>n = 666 | Test statistics | Significance<br>(p value) |
|---|---|---|---|---|---|
| Smoking status, N<br>(never: current: past smoker) | 658: 167: 319 | 265: 68: 145 | 393: 99: 174 | $X^2 (2) = 2.46$ | 0.292 |
| Hypertension, N (yes: no) | 58: 1086 | 37: 441 | 21: 645 | $X^2 (1) = 11.23$ | **<0.001** |
| Age, years (±SD) | 43.99 (± 5.51) | 44.3 (±5.35) | 43.77 (±5.61) | $t (1057) = 1.53$ | 0.1449 |
| BMI, kg/m2 (±SD) | 24.72 (±3.94) | 26.17 (±3.41) | 23.7 (±3.96) | $t (1079) = 10.71$ | **<0.001** |
| Hospitalization 01, N (yes: no) | 0:10 | 0:4 | 0:6 | Fisher's exact test | 1 |
| Hospitalization 02, N (yes: no) | 11:113 | 7:44 | 4:69 | $X^2 (1) = 2.52$ | 0.2047 |
| Hospitalization 03, N (yes: no) | 2:23 | 0:12 | 2:11 | $X^2 (1) = 0.46$ | 0.4973 |
| Hospitalization 04, N (yes: no) | 3:47 | 0:24 | 3:23 | $X^2 (1) = 1.25$ | 0.2625 |

Baseline characteristics stratified according to sex were collected by questionnaires completed within the GAPP study. Information about hospitalization was collected four times (01–04) in the scope of the study centre visit for SARS-CoV-2 antibody tests. It was a part of the questionnaire for SARS-CoV-2 positive participants and represented a measure of disease severity. Data are presented as mean ± SD or number. The test statistic and the corresponding p-value are shown for the comparison between the female and male group for each variable.

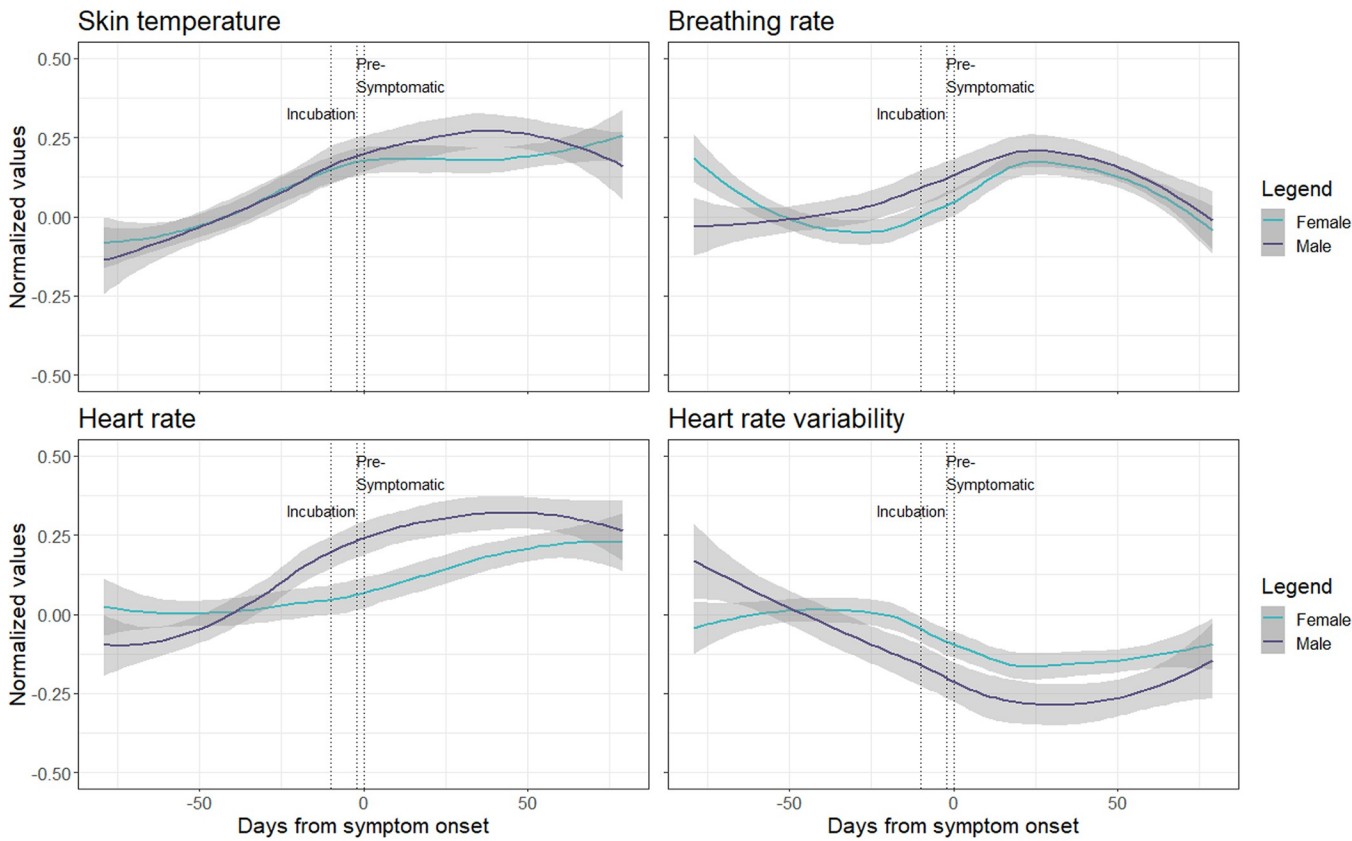

**Fig 2. Trajectory of the four analysed physiological parameters across the course of a confirmed COVID-19 infection centred around participant-reported symptom onset.** The values of each physiological parameter (with 95% CIs) were normalized according to each individual's baseline measurements and collapsed across females (n = 56) and males (n = 26).

**Table 2. Results from multilevel linear mixed models showing the main effects of infection phase and sex as well as the interactions between the two with regards to changes in physiological signals.**

|  | Skin temperature (degree Celsius) | Breathing rate (breaths per minute) | Heart rate (beats per minute) | Heart rate variability |
|---|---|---|---|---|
| Intercept | **35.01 (<0.0001)** | **13.51 (<0.0001)** | **46.98 (<0.0001)** | **4.3 (<0.0001)** |
| Infection phase |  |  |  |  |
| Baseline | Reference | Reference | Reference | Reference |
| Incubation | 0.18 (0.15) | 0.33 (0.13) | 1.49 (0.12) | -0.25 (0.12) |
| Pre-symptomatic | 0.23 (0.26) | 0.71 (0.17) | 1.26 (0.41) | -0.18 (0.42) |
| Symptomatic | **0.74 (<0.0001)** | **2.93 (<0.0001)** | **6.88 (<0.0001)** | **-0.93 (<0.0001)** |
| Recovery | **0.22 (0.0006)** | **0.38 (0.004)** | **2.17 (0.003)** | -0.28 (0.09) |
| Sex, female | **0.45 (<0.0001)** | 0.91 (0.06) | **4.96 (0.001)** | **-1.35 (<0.0001)** |
| Interaction |  |  |  |  |
| Sex*Incubation | -0.02 (0.74) | -0.2 (0.11) | -0.36 (0.5) | 0.09 (0.34) |
| Sex*Pre-symptomatic | -0.01 (0.92) | -0.26 (0.38) | 0.07 (0.93) | 0.04 (0.78) |
| Sex*Symptomatic | **-0.28 (<0.0001)** | **-1.31 (<0.0001)** | **-3.09 (0.0001)** | **0.43 (<0.0001)** |
| Sex*Recovery | -0.04 (0.23) | **-0.25 (0.001)** | **-0.96 (0.02)** | 0.11 (0.25) |

Unstandardized beta coefficients are presented, with *p*-values in parentheses and in bold if lower than 0.05. Sex was coded such that positive coefficients represent larger values in females.

COVID-19 infection over and above changes associated with participant age, BMI, hypertension, medication, alcohol, and recreational drugs. These variables were selected based on previously reported associations with physiological signals [35]. Furthermore, we observed sex differences in BMI and hypertension in the current sample (Table 1) and, therefore, examined in the sensitivity analysis whether these effects can account for the sex differences found in the main analysis. Hypertension, medication and recreational drug intake were binary variables (i.e., yes/no), while alcohol intake was represented through four categories (i.e., none/1-2 drinks/3-4 drinks/5+ drinks with none as the reference category). As outlined in the main analysis, four multilevel models were computed (i.e., one for each physiological parameter) additionally including the described variables as fixed effects. In these models, the interactions between sex and phase of infection remained unchanged, indicating that they cannot be explained by the influence of the added variables (S1 Table).

## Sex-specific differences in algorithm's performance

Table 3 provides a by-sex breakdown of the algorithm's performance. Sensitivity score can be found as the recall of the positive class (days with an existent SARS-CoV2 infection), while

**Table 3. Performance metrics of the machine learning algorithm for female and male participants.**

| Participant Sex | Class | Precision | Recall | F-score |
|---|---|---|---|---|
| All | 0 | 12.36 | 68.421 | 19.048 |
|  | 1 | 91.599 | 41.509 | 78.331 |
| Female | 0 | 12.977 | 60.69 | 20.859 |
|  | 1 | 92.147 | 53.125 | 73.181 |
| Male | 0 | 10.811 | 80.0 | 15.385 |
|  | 1 | 92.308 | 26.667 | 85.714 |

Sensitivity score can be found as the recall of the positive class (i.e., days with an existent SARS-CoV2 infection), while specificity is the recall of the negative class (i.e., days without a SARS-CoV2 infection).

**Table 4. SARS-CoV-2 Nucleocapsid (N) antibody (AB) values stratified according to sex.**

| Variables | Male (n = 7) | Female (n = 7) | Test statistics | Significance (p value) |
|---|---|---|---|---|
| SARS-CoV-2 N AB run1 | 17.7 (5.8–83.5) | 54.8 (5.7–135.1) | W = 13 | 0.14 |
| | Male (n = 51) | Female (n = 68) | | |
| SARS-CoV-2 N AB run2 | 34.1 (1–183.7) | 33.4 (1.7–212.2) | W = 1753 | 0.92 |
| | Male (n = 62) | Female (n = 85) | | |
| SARS-CoV-2 N AB run3 | 40.05 (1–274) | 29.7 (1.4–234.3) | W = 2772 | 0.59 |
| | Male (n = 76) | Female (n = 102) | | |
| SARS-CoV-2 N AB run4 | 17.95 (1.2–221) | 36.59 (1–266.4) | W = 4280 | 0.235 |

Data are presented as median and interquartile range.

specificity is the recall of the negative class (days without a SARS-CoV2 infection). The algorithm showed the same precision (i.e., 92) when giving a SARS-CoV2 positive alert across participant sex. Cross-class recall was more balanced among females than males in our sample. Detecting 53% of SARS-CoV-2 positive days in females, the algorithm performed less well in males (26% of SARS-CoV2 positive cases detected).

### Sex-specific differences in antibody titres of SARS-CoV-2 Nucleocapsid after COVID-19 infection

Antibody titres of the female and male sub-groups were not significantly different across runs. Nucleocapsid antibody values in run 1 trended higher in female participants (Table 4).

## Discussion

The presented study examined sex-specific differences in physiological parameters among 82 individuals with a documented SARS-CoV-2 infection. We found that male participants experienced significantly larger increases in wrist skin temperature, breathing rate and heart rate as well as larger decreases in heart rate variability during the symptomatic period compared to females. In one of the first prospective cohort studies relying on wearable sensor technology to collect real-time continuous physiological signals, we provide evidence for sex-based differential physiological responses to COVID-19.

Considering the higher mortality and hospitalization rates observed in male COVID-19 patients [9], our findings may reflect sex-specific biological responses to the infection. In line with previous work [16], we did not observe any differences between the sexes with regard to antibody titers. However, Takahashi et al. [16] observed a stronger acute T-cell response in females as compared to male COVID-19 patients. The poorer T-cell response in men was associated with their worse disease progression. On the other hand, the authors measured higher levels of several pro-inflammatory innate immunity chemokines and cytokines in men as compared to women. They thus concluded that the early phase of COVID-19 is associated with key sex differences in immunological mechanisms potentially accounting for the differential disease progression between women and men.

Given that the sex differences in physiological signals in our study are most pronounced during the symptomatic phase, we propose that they reflect the above-mentioned sex-specific immunological mechanisms [36]. Inflammatory markers (e.g., cytokines) have been shown to reflect disease severity in COVID-19 [37]. As the autonomic nervous system is known to modulate inflammation [38] and the examined physiological signals reflect the function of the autonomic nervous system [39], our findings suggest support for differential immunological responses to COVID-19 between the sexes.

Importantly, altered physiological signals such as decreased heart rate variability and increased skin temperature have been proposed as prognostic markers for several disorders, including cardiovascular disease [40] as well as infectious diseases like COVID-19 [18,41–43]. Modern wearable technology represents a unique and powerful framework to collect continuous real-time physiological data. The predictive value of physiological signals combined with the reliable history of measurements provided by wearables opens up new avenues to inform clinical actions and support future precision medicine approaches incorporating a variety of individual factors into clinical decisions (reviewed in Mitratza et al. [44]).

An important step towards precision medicine can be made by considering sex differences in modern digital health solutions. Historically, women have been underrepresented in clinical trials, leading to medical solutions focusing on men at the risk to women's health [45]. Many diseases differ between female and male patients with regard to the prevalence, progression, or response to treatment [46]. For example, minor stroke is more often missed in female than male [47] patients, possibly due to definitions in clinical diagnosis reflecting typical manifestations in males [43]. More recently, a sex bias has been recognized in modern ML solutions that are often developed and trained on male data and thus result in better performance in men [48]. Therefore, in the presented work, we examined sex differences in the performance of our ML algorithm for early detection of COVID-19. The algorithm reached a higher sensitivity for female participants. We postulate this difference may be due to the larger sample size in the female group. However, the algorithm's precision was the same in both groups, indicating that it is suitable for use in both men and women, as intended.

## Limitations

While our study belongs to the first research to consider sex-based differences in COVID-19 detection using digital health, future work could continue to build upon our findings by examining the casual mechanism underlying differences between SARS-Cov-2 infected men and women. In particular, the inability to disentangle immunological versus menstrual-driven changes in physiological parameters among female participants limits our research's generalizability. In menstruating women, a specific pattern has been recognized in the trajectory of physiological signals across the menstrual cycle, mirroring cycle-based shifts in sex hormones [27]. Particularly during the follicular phase of the menstrual cycle, decreased skin temperature, heart rate and breathing rate have been observed, while heart rate variability was increased. In contrast, the luteal phase was associated with increases in skin temperature, heart rate and breathing rate as well as decreases in heart rate variability, corresponding to the pattern found in COVID-19 patients. Sex differences in physiological signals measured in the current study may thus partly be due to hormonal impact. We cannot exclude such influence as we had limited information about female participants' menstrual cycle or reproductive health (e.g., usage of hormonal birth control menopausal status). Future researchers may wish to record participants' menstrual status and measure hormone levels directly, to probe the relationship between sex hormones and physiological differences.

Nevertheless, we believe that menses-driven changes in physiology do not adequately explain the sex differences in our results, as the dynamics of the observed physiological signals are in line with previous reports regarding COVID-19 and include increased skin temperature, heart rate and breathing rate as well as decreased heart rate variability during infection [20]. Additionally, the most pronounced sex differences in our study occurred during the symptomatic period, suggesting a disease-triggered disparity among males and females. Furthermore, hormonal influence offers a plausible explanation only in the first half of the menstrual cycle. The physiological changes observed in its second half could only amplify the trajectory found

during COVID-19 in females and thus mask the sex differences in our study. Moreover, the magnitude of physiological changes during COVID-19's symptomatic phase in the current study is, for all parameters, more than twice as large as the previously reported magnitude of changes across the menstrual cycle [27]. For example, we found that skin temperature increases by 0.7 degrees during the symptomatic phase of COVID-19, whereas this measure's largest increase during the menstrual cycle is 0.2 degrees during the late luteal phase [27]. Of note, 30% (n = 17) of females in the sample were older than 45 years; peri- or post-menopausal, they were beyond natural reproductive age and thus unlikely to experience menses-modulating effects on their physiological parameters. Finally, we do not expect that the distribution of menstrual cycle phases follows a specific pattern for our participants (e.g., in complete synchronicity); rather, we expect each eumenorrheic woman to cycle on her own timeline and the alignment of menstrual phases between participants to occur at random. Taken together, we believe that the hormonal impact on our findings is minimal.

Another limitation important to note is the potential effect of recall bias on our findings. The COVID-19 symptom onset date was determined based on the participants' retrospective reports, and the classification of the relevant infection periods (i.e., incubation, pre-symptomatic and symptomatic period) was based on this date. Therefore, an unreliable report would be associated with an inaccurate definition of the infection periods leading to shifts in trajectories of physiological signals. Furthermore, in the effort to smooth the data in the model, the abrupt changes in physiological signals after infection generated gradual alterations in the estimated trajectory. The deviations from the baseline during the first and last days may be reflective of such model artifacts (Fig 2). Finally, it is important to note that we did not adjust any parameters from our statistical tests to account for multiple testing. Therefore, we acknowledge chances for type 1 error in our findings. Nevertheless, we believe that our research provides important initial insights to be confirmed in future investigations. Furthermore, upcoming research should explore the mechanisms behind these sex differences, including the roles of sex hormones, genetic factors, and immune responses. Finally, the development of sex-specific treatment strategies, leveraging the insights gained from our study, holds potential for improved patient care and outcomes.

## Conclusion

Our study demonstrates sex differences in physiological responses to COVID-19. The results highlight the importance of taking sex into account in medical treatment and care of COVID-19 patients, as well as when validating infection detection algorithms in digital health. Moreover, we reveal the potential of continuous real-time physiological signals as a clinical tool to inform future precision medicine approaches. Wearable technology, capable of providing a reliable history of measurements, can empower clinicians with invaluable insights into individual patient health, enabling more personalized and timely interventions that hold promise for improved patient outcomes in the fight against COVID-19 and beyond.

## Supporting information

**S1 Table. Results from multilevel linear mixed models showing the main effects of infection phase, sex, age, medication, drug and alcohol intake, BMI, and hypertension, as well as interactions between sex and infection phase with regards to changes in physiological signals.**
(DOCX)

## Acknowledgments

We thank the GAPP participants who enrolled in this study. Additionally, the authors thank the following for their contributions to the study: The local study team in Vaduz, FL, the different teams at the Dr Risch medical laboratories in Vaduz and Buchs, CH. We would also like to thank the Coobx AG in Balzers, FL, for the provision of 3D printed bracelet extensions for persons with large wrists. Addressing data protection issues, we acknowledge the substantial collaborative support of the Elleta AG as well as the national data protection agency in Liechtenstein. We thank the government of the Principality of Liechtenstein, the health ministers, and the Liechtenstein Office of Public Health for their support. Finally, our thanks are especially due to the Princely House of Liechtenstein, which gave decisive support that enabled the initiation of this project.

**List of the members of the COVID-19 remote early detection (COVID-RED) consortium**

\* **Lorenz Risch** is lead author of this group (lorenz.risch@risch.ch)

| First name | Surname | Organization |
| --- | --- | --- |
| Andjela | Markovic | Ava AG, Switzerland |
| Maja | Rudinac | Ava AG, Switzerland |
| Maureen | Cronin | Ava AG, Switzerland |
| Vladimir | Kovacevic | Ava AG, Switzerland |
| Kirsten | Grossmann | Dr. Risch Anstalt, Liechtenstein |
| **Lorenz** | **Risch\*** | Dr. Risch Anstalt, Liechtenstein |
| Martin | Risch | Dr. Risch Anstalt, Liechtenstein |
| Ornella | Weideli | Dr. Risch Anstalt, Liechtenstein |
| Billy | Franks | Julius Clinical, The Netherlands |
| Brianna | Goodale | Julius Clinical, The Netherlands |
| Ellen | Dutman | Julius Clinical, The Netherlands |
| Eric | Houtman | Julius Clinical, The Netherlands |
| Glenn | Van Wigcheren | Julius Clinical, The Netherlands |
| Hans | Van Dijk | Julius Clinical, The Netherlands |
| Ishak | Elmouhajir | Julius Clinical, The Netherlands |
| Jeffrey | Burggraaff | Julius Clinical, The Netherlands |
| Jon | Bouwman | Julius Clinical, The Netherlands |
| José | Broersen | Julius Clinical, The Netherlands |
| Jungyeon | Choi | Julius Clinical, The Netherlands |
| Kai | Hage | Julius Clinical, The Netherlands |
| Lotte | Smets | Julius Clinical, The Netherlands |
| Maartje | Hoffmann | Julius Clinical, The Netherlands |
| Marcel | van Willigen | Julius Clinical, The Netherlands |
| Marjolein | Jansen | Julius Clinical, The Netherlands |
| Myrna | Verhulst | Julius Clinical, The Netherlands |
| Niki | de Vink | Julius Clinical, The Netherlands |
| Paul | Klaver | Julius Clinical, The Netherlands |
| Pieter | van der Meer | Julius Clinical, The Netherlands |
| Tessa | Heikamp | Julius Clinical, The Netherlands |
| Timo | Brakenhoff | Julius Clinical, The Netherlands |
| Titia | Leurink | Julius Clinical, The Netherlands |
| Wendy | van Scherpenzeel | Julius Clinical, The Netherlands |

(*Continued*)

| Wout | Aarts | Julius Clinical, The Netherlands |
|------|-------|----------------------------------|
| Alison | Kuchta | Roche, The Netherlands |
| Christian | Simon | Roche, The Netherlands |
| Santiago | Montes | Roche, The Netherlands |
| Aren | Boogaard | Sanquin, The Netherlands |
| Florine | van Milligen | Sanquin, The Netherlands |
| Floris | Loeff | Sanquin, The Netherlands |
| Jim | Keijser | Sanquin, The Netherlands |
| Lea | Berkhout | Sanquin, The Netherlands |
| Maurice | Steenhuis | Sanquin, The Netherlands |
| Nadine | Commandeur | Sanquin, The Netherlands |
| Olvi | Christianawati | Sanquin, The Netherlands |
| Sofie | Keijzer | Sanquin, The Netherlands |
| Theo | Rispens | Sanquin, The Netherlands |
| Ariel | Dowling | Takeda, USA |
| Steve | Emby | Takeda, USA |
| Charisma | Hehakaya | University Medical Center Utrecht, The Netherlands |
| Daniel | Oberski | University Medical Center Utrecht, The Netherlands |
| George | Downward | University Medical Center Utrecht, The Netherlands |
| Gulseren | Yalvac | University Medical Center Utrecht, The Netherlands |
| Hans | Reitsma | University Medical Center Utrecht, The Netherlands |
| Janneke | Wijgert, van de | University Medical Center Utrecht, The Netherlands |
| Marianna | Mitratza | University Medical Center Utrecht, The Netherlands |
| Nathalie | Vigot | University Medical Center Utrecht, The Netherlands |
| Patricia | Bruijning | University Medical Center Utrecht, The Netherlands |
| Pieter | Stolk | University Medical Center Utrecht, The Netherlands |
| Rick | Grobbee | University Medical Center Utrecht, The Netherlands |
| Amos | Folarin | University College London, UK |
| Johann | Fevrier | University College London, UK |
| Pablo | Fernandez Medina | University College London, UK |
| Richard | Dobson | University College London, UK |
| Spiros | Denaxas | University College London, UK |
| Eskild | Fredslund | VIVE, Denmark |
| Jesper | Strømstad | VIVE, Denmark |
| Serkan | Korkmaz | VIVE, Denmark |

## Author Contributions

**Conceptualization:** Martin Risch, Harald Renz, Raphael Twerenbold, Paul Klaver, Timo B. Brakenhoff, Billy Franks, Marianna Mitratza, George S. Downward, Ariel Dowling, Santiago Montes, Diederick E. Grobbee, Maureen Cronin, David Conen, Brianna M. Goodale, Lorenz Risch.

**Data curation:** Kirsten Grossmann, Andjela Markovic, Stefanie Aeschbacher, Martina Rothenbühler, Daniel Leibovitz, Vladimir Kovacevic, Brianna M. Goodale.

**Formal analysis:** Kirsten Grossmann, Martin Risch, Andjela Markovic, Ornella C. Weideli, Nadia Wohlwend, Martina Rothenbühler, Brianna M. Goodale.

**Funding acquisition:** Martin Risch, Diederick E. Grobbee, Maureen Cronin, David Conen, Lorenz Risch.

**Investigation:** Kirsten Grossmann, Stefanie Aeschbacher, Ornella C. Weideli, Marianna Mitratza.

**Methodology:** Kirsten Grossmann, Martin Risch, Andjela Markovic, Paul Klaver, Timo B. Brakenhoff, Billy Franks, George S. Downward, Santiago Montes, Duco Veen, Diederick E. Grobbee, Maureen Cronin, David Conen, Brianna M. Goodale, Lorenz Risch.

**Project administration:** Kirsten Grossmann, Ornella C. Weideli, Thomas Lung, Harald Renz, Santiago Montes, Diederick E. Grobbee, Maureen Cronin.

**Resources:** Kirsten Grossmann, Martin Risch, Stefanie Aeschbacher, Ornella C. Weideli, Marc Kovac, Nadia Wohlwend, Corina Risch, Dorothea Hillmann.

**Supervision:** Martin Risch, Stefanie Aeschbacher, Raphael Twerenbold, Diederick E. Grobbee, Maureen Cronin, David Conen, Brianna M. Goodale, Lorenz Risch.

**Validation:** Kirsten Grossmann, Andjela Markovic, Ornella C. Weideli, Nadia Wohlwend, Corina Risch, Dorothea Hillmann, Martina Rothenbühler, Daniel Leibovitz, Vladimir Kovacevic, Duco Veen.

**Visualization:** Kirsten Grossmann, Andjela Markovic, Fiona Pereira.

**Writing – original draft:** Kirsten Grossmann, Martin Risch, Andjela Markovic, Fiona Pereira, Maureen Cronin, David Conen, Brianna M. Goodale, Lorenz Risch.

**Writing – review & editing:** Kirsten Grossmann, Martin Risch, Andjela Markovic, Stefanie Aeschbacher, Ornella C. Weideli, Laura Velez, Marc Kovac, Fiona Pereira, Nadia Wohlwend, Corina Risch, Dorothea Hillmann, Thomas Lung, Harald Renz, Raphael Twerenbold, Martina Rothenbühler, Daniel Leibovitz, Vladimir Kovacevic, Paul Klaver, Timo B. Brakenhoff, Billy Franks, Marianna Mitratza, George S. Downward, Ariel Dowling, Santiago Montes, Duco Veen, Diederick E. Grobbee, Maureen Cronin, David Conen, Brianna M. Goodale, Lorenz Risch.

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
