## [Decision Letter · Decision Letter 0]

9 Oct 2023

PONE-D-23-28587Sex-specific differences in physiological parameters related to SARS-CoV-2 infections among a national cohort (COVI-GAPP study)PLOS ONE

Dear Dr. Risch,

Thank you for submitting your manuscript to PLOS ONE. After careful consideration, we feel that it has merit but does not fully meet PLOS ONE’s publication criteria as it currently stands. Therefore, we invite you to submit a revised version of the manuscript that addresses the points raised during the review process.

We look forward to receiving your revised manuscript.

Kind regards,

Ramada Rateb Khasawneh

Academic Editor

PLOS ONE

[I have read the journal's policy and the authors of this manuscript have the following competing interests: Lorenz Risch, and Martin Risch are key shareholders of the Dr Risch Medical Laboratory. David Conen has received consulting fees from Roche Diagnostics, outside of the current work. Andjela Markovic, Vladimir Kovacevic and Brianna Goodale are past employees of Ava AG. Billy Franks is a former employee of the Julius Clinic and now employee of Haleon. The other authors have declared that no competing interests exist.]. 

4. Please amend the manuscript submission data (via Edit Submission) to include author Duco Veen.

5. One of the noted authors is a consortium [COVID-RED consortium]. In addition to naming the author group, please list the individual authors and affiliations within this group in the acknowledgments section of your manuscript. Please also indicate clearly a lead author for this group along with a contact email address.

6. We note that Figure 1 in your submission contain copyrighted images. All PLOS content is published under the Creative Commons Attribution License (CC BY 4.0), which means that the manuscript, images, and Supporting Information files will be freely available online, and any third party is permitted to access, download, copy, distribute, and use these materials in any way, even commercially, with proper attribution. For more information, see our copyright guidelines: http://journals.plos.org/plosone/s/licenses-and-copyright.

A. You may seek permission from the original copyright holder of Figure 1 to publish the content specifically under the CC BY 4.0 license. 

B. If you are unable to obtain permission from the original copyright holder to publish these figures under the CC BY 4.0 license or if the copyright holder’s requirements are incompatible with the CC BY 4.0 license, please either i) remove the figure or ii) supply a replacement figure that complies with the CC BY 4.0 license. Please check copyright information on all replacement figures and update the figure caption with source information. If applicable, please specify in the figure caption text when a figure is similar but not identical to the original image and is therefore for illustrative purposes only.

Additional Editor Comments:

the paper is a good paper, but may I ask you to submit the row data 

Reviewers' comments:

Reviewer's Responses to Questions

**Comments to the Author**

1. Is the manuscript technically sound, and do the data support the conclusions?

Reviewer #1: Yes

Reviewer #2: Yes

Reviewer #3: Yes

2. Has the statistical analysis been performed appropriately and rigorously? 

Reviewer #1: Yes

Reviewer #2: Yes

Reviewer #3: Yes

3. Have the authors made all data underlying the findings in their manuscript fully available?

Reviewer #1: Yes

Reviewer #2: No

Reviewer #3: No

4. Is the manuscript presented in an intelligible fashion and written in standard English?

Reviewer #1: Yes

Reviewer #2: Yes

Reviewer #3: Yes

5. Review Comments to the Author

Reviewer #1: Overall, the study is very well organised and provides a clear and comprehensive overview of the background, research gap, and objectives of the study. The results are clearly stated and discussed. However, I have added a few suggestive comments along with the original manuscript file to be addressed for clarity. I suggested few grammatical and sentence composition suggestions as well using tracked changes, the authors can approve if they find appealing. Please refer to the original file for the comments.

Reviewer #2: 1. I would suggest removing funding sources from the Abstract section.

2. Table 4, first row, looking at the mean and SD of 32.66 (±34.74), clearly it doesn't follow the Gaussian distribution. Consider representing the data in median and IQR as the others.

Reviewer #3: The article followed the right research and Public ethics. But my question: Is the underlying cardiovascular conditions of the patients play a role in generating Sex-specific differences in physiological parameters relating to the SARS-CoV-2 infections

among a research participants? I need some explanation on this

6. PLOS authors have the option to publish the peer review history of their article (what does this mean?). If published, this will include your full peer review and any attached files.

Reviewer #1: No

Reviewer #2: **Yes: **Dr. Abdurrahman Ahmad El-fulaty

Reviewer #3: **Yes: **Emmanuel Adamolekun

---

## [Author Response · Author response to Decision Letter 0]

18 Dec 2023

See document 'Response to Reviewers_20231123' for our answers to the reviewers.

---

## [Decision Letter · Decision Letter 1]

5 Jan 2024

Sex-specific differences in physiological parameters related to SARS-CoV-2 infections among a national cohort (COVI-GAPP study)

PONE-D-23-28587R1

Dear Dr. Risch,

We’re pleased to inform you that your manuscript has been judged scientifically suitable for publication and will be formally accepted for publication once it meets all outstanding technical requirements.

Kind regards,

Ramada Rateb Khasawneh

Academic Editor

PLOS ONE

Additional Editor Comments (optional):

Nice Paper ......Good Luck

Reviewers' comments:

Reviewer's Responses to Questions

**Comments to the Author**

1. If the authors have adequately addressed your comments raised in a previous round of review and you feel that this manuscript is now acceptable for publication, you may indicate that here to bypass the “Comments to the Author” section, enter your conflict of interest statement in the “Confidential to Editor” section, and submit your "Accept" recommendation.

Reviewer #1: All comments have been addressed

Reviewer #2: All comments have been addressed

2. Is the manuscript technically sound, and do the data support the conclusions?

Reviewer #1: Yes

Reviewer #2: Yes

3. Has the statistical analysis been performed appropriately and rigorously? 

Reviewer #1: N/A

Reviewer #2: Yes

4. Have the authors made all data underlying the findings in their manuscript fully available?

Reviewer #1: Yes

Reviewer #2: (No Response)

5. Is the manuscript presented in an intelligible fashion and written in standard English?

Reviewer #1: Yes

Reviewer #2: Yes

6. Review Comments to the Author

Reviewer #1: (No Response)

Reviewer #2: (No Response)

7. PLOS authors have the option to publish the peer review history of their article (what does this mean?). If published, this will include your full peer review and any attached files.

Reviewer #1: No

Reviewer #2: **Yes: **Dr. Abdurrahman Ahmad El-fulaty

---

## [Editor Report · Acceptance letter]

14 Feb 2024

PONE-D-23-28587R1 

PLOS ONE

Dear Dr. Risch, 

I'm pleased to inform you that your manuscript has been deemed suitable for publication in PLOS ONE. Congratulations! Your manuscript is now being handed over to our production team.

Kind regards, 

on behalf of

Dr. Ramada Rateb Khasawneh 

Academic Editor

PLOS ONE